# The Influence of Parental Environmental Exposure and Nutrient Restriction on the Early Life of Offspring Growth in Gambia—A Pilot Study

**DOI:** 10.3390/ijerph192013045

**Published:** 2022-10-11

**Authors:** Ousman Bajinka, Amadou Barrow, Sang Mendy, Binta J. J. Jallow, Jarry Jallow, Sulayman Barrow, Ousman Bah, Saikou Camara, Modou Lamin Colley, Sankung Nyabally, Amie N. Joof, Mingming Qi, Yurong Tan

**Affiliations:** 1Department of Medical Microbiology, Central South University, Changsha 410078, China; 2China-Africa Research Center of Infectious Diseases, School of Basic and Medical Sciences, Central South University, Changsha 410078, China; 3School of Medicine and Allied Health Sciences, University of The Gambia, Kanifing 3530, The Gambia; 4Ministry of Health, Banjul P.O. Box 273, The Gambia; 5Department of Obstetrics, Zhuzhou Hospital Affiliated to Xiangya School of Medicine, Central South University, Changsha 410017, China

**Keywords:** epigenetics, toxicant exposure, maternal stress, parental smoking, offspring health

## Abstract

Background: The role of the germline in epigenetic transgenerational inheritance starts with environmental factors, acting on the first generation of a gestating mother. These factors influence the developing second-generation fetus by altering gonadal development, thereby reprogramming the primordial germ cell DNA methylation and leading to consequences that might be seen along generations. Objective: Despite these epigenetic factors now surfacing, the few available studies are on animal-based experiments, and conducting a follow-up on human intergenerational trials might take decades. To this response, this study aimed to determine the influence of parental energy, toxicant exposure, age, and nutrient restriction on the early life of offspring growth in Gambia. Method: This pilot study was based on population observation and combined both maternal and paternal factors across the country between August and October 2021. It captures the lifestyle and health detailed account of 339 reproductive parents and their last born (child under 5 years) using a structured interview questionnaire performed by nurses and public health officers. Results: This study showed that parents who worked in industrial areas were more likely to have offspring with poor psychosocial skills. In addition, mothers who are exposed to oxidative stress and high temperatures are more likely to have offspring with poor psychosocial skills. Mothers who consume a high-protein diet were almost three times more likely to have infants with good psychosocial skills in their offspring. Furthermore, there was a negative correlation between maternal stress during pregnancy and the psychosocial skills of offspring. Conclusion: This study was able to ascertain if the maternal diet during gestation, toxicant exposure, maternal stress, and parental smoking habits have an influence on the early life of offspring. While the study recommends a large sample size study to eliminate selection bias, there should be an increased level of awareness of mothers of their offspring’s health and their husbands’ lifestyles that might influence the adulthood health of their offspring.

## 1. Introduction

In addition to genetic influence, the environment can determine the health and well-being of progeny. A typical example is seen between identical twins whose DNA is very similar but has different levels of risk to various diseases due to environmental exposures [1]. While this influence is not entirely from DNA as spontaneous genetic mutations, epigenetic changes include alterations in DNA methylation status, the post-translational modification of histones, and non-protein-coding RNA [2]. In addition to the genetic constitutions, which are DNA inherited by an organism, the trajectory in human development that led to the mature phenotype is also determined by mechanisms acting during critical windows in early life. This influence was studied to establish stable patterns of gene expression [3]. Although this is not accounted for by variations in DNA sequence, it causes permanent phenotypic consequences for offspring such as implications for psychological and developmental problems, stress response, and immune functioning [4]. Furthermore, unlike genomics, epigenetic marks are reversible in nature. In a sense, multiple rounds of epigenetic reprogramming can completely erase the existing patterns. How these affect an individual’s germline and how it can be transmitted to the preceding generations are interesting questions genetics are yet to unearth. This approach entails the combination of genetic and epigenetic modifications and the mechanism involved in determining the phenotypes of individuals and their offspring [2,5].

The germline, which alters DNA methylation, will become permanently programmed. Consequently, this imprint-like gene will be passed down through the germline to subsequent generations [1,2]. The developing embryo generated from this specific germline will be an altered epigenome, which may affect developing somatic cells and tissues. The integrated epigenetic modifications into the genome of individuals have the potential to modulate gene expression. In addition, it can modulate gene activity in enhancer and promoter domains [2]. Similarly, genetic mutations alter sequence availability for methylation and histone binding. These combinations are studied to present a stable inheritance feature for the next generation or germline. These alterations to the nuclear composition are due to environmental factors, ageing, diet, and toxicant exposure [2]. Cytochrome P450 activity, gestational duration, maternal glucose, immune functions, smaller head circumference, and the likely maternal blood pressure are all crucial for fetal growth and low and high birth weights to be specific [6]. Moreover, maternal blood pressure linked to offspring’s low birth weight is strongly evident in genetics instead of an adverse intrauterine environment [7,8]. Prenatal exposure to an earthquake results in earlier delivery in addition to the reduced length and head circumference, and this is relative to the trimester of exposure [9]

These risk factors for epigenetic changes are maternal undernutrition, hyperglycemia, birth weight, obesity, a high-fat and low-protein diet, and diabetes mellitus. Furthermore, advanced age, smoking, and environmental chemical exposure affect the offspring’s metabolic and cardiovascular health later in life [10]. The emerging associating factors are paternal obesity, nutritional habits, diabetes mellitus status, advanced age, and exposure to environmental chemicals or cigarette smokers. These are seen with episodes of adverse effects on metabolic and cardiovascular health in offspring [11]. Environmental chemicals alter hormone signaling or disrupt hormone production. This causes endocrine disruption, which has profound consequences on hormones’ role in human development [1]. Nephrogenesis is impacted by poor placental function, inadequate diet, maternal stress, maternal smoking, and alcohol consumption. A reduced nephron endowment with a high risk of developing hypertension and chronic kidney disease (CKD) may result [12]. In addition to maternal nutrition, paternal periconceptional nutrition affects offspring’s likelihood of developing chronic metabolic-related conditions and cardiovascular diseases due to epigenetic imprinting [13,14]. Moreover, maternal obesity affects the cognitive function and mental health of the offspring [15].

It is obvious that an improved diet plays a big role in general health and wellbeing. This is inconsistent with proper maternal nutrition during pregnancy. Monitoring the glycemic level will add up to maternal health and reduce the risk of later obesity in infants and body composition [16]. For instance, around conception and in early pregnancy, insulin resistance and glycemia affect fetal nutrient supply. Moreover, throughout the gestation period, maternal-feto-placental communications will be affected. All these will be reflected in later postnatal health [16,17]. Following the available nutritive diet during the period of early gestation through lactation, this study would be able to ascertain the evidence of being exposed in utero to famine and a greater risk of poor health in later life [18]. In response to these epigenetic imprints, this study is designed to determine the influence of parental energy, toxicant exposure, age, and nutrient restriction on the early life of offspring growth in Gambia.

## 2. Methodology

### 2.1. Study Design

The study was a population-based analytical cross-sectional study on parental energy influence, toxicant exposure, age, and nutrient restriction on offspring growth. This observational study is associated with the effect of parental exposure to energy and nutrient restrictions in utero on their children’s growth in rural Gambia [17]. While a previous study investigated matriline influences on fetal growth and postnatal growth under patriline intergenerational influences in only one sub-district in Gambia, the current study is designed to combine both maternal and paternal factors across the country.

### 2.2. Study Setting and Duration

This study was conducted between August and October 2021 in Gambia. Gambia is the smallest West African nation in mainland Africa, located on the western coast of Africa. It is long and narrow in shape, extending 487 km into the hinterlands, with an average width of 24 km. At the point where the River Gambia meets the Atlantic Ocean, the width of the country is twice the average at more than 48 km. The country is bound on three sides by Senegal and on the west by the Atlantic Ocean. The population of Gambia is estimated to be 2.4 million [19].

Study population: This survey includes lifestyles and health details for reproductive mothers and their last born (child under 5 years).

### 2.3. Study Variables

Inclusion and exclusion criteria: Mothers with their last born at least under 5 years from all provinces across the Gambia met the inclusion criteria. However, mothers who were recruited to take supplements during pregnancy were excluded from this study.

### 2.4. Sample Size

The minimum sample size was determined using a single population proportion formula; n = (Zα/2)^2^ × p (1 − p)/d^2^), where n = the required sample size for this study; Zα/2(1.96) = the significance level at α = 0.05 with a 95% confidence interval; p = the proportion of reproductive women with a child under 5 years of age, which is estimated to be 25.2%; and d = the margin of error (5%). Upon inserting values into the equation, the minimum sample size was 293. We finally recruited 339 respondents that include the 10% non-response rate and missing and incomplete responses.

### 2.5. Data Collection Tools

A structured interview questionnaire was developed and pretested by the research team. During the pretesting stage of the tools, redesigning of some sections that were found to be ambiguous, misleading, and culturally sensitive introspectively, etc., were revised and adjusted in the validated version. These enhance the tool’s validity and reliability with a Cronbach Alpha score of 84%. The tool has major sections, which include socio-demographic information, the mother’s health profile, etc.

### 2.6. Data Collection Procedure

The interviews were conducted by trained public health officers, trained nurses, and some student nurses, and the information was submitted online simultaneously. The authentication of some information was based on the consultation of the ANC card of the child in question. Regarding the geographical information system, the study was able to track the map as to how and where the data in relation to the participants were collected. The research team included public health professionals, nurses, and medical students from the University of Gambia, who conducted interviews in-house visits while assessing the antenatal care (ANC) cards to supplement the information.

### 2.7. Data Quality and Bias Mitigation

The questionnaires were pretested with approximately 5% of the estimated sample to check for validity and reliability. The study questionnaires were administered by the research team. To avoid potential information bias, the study details, including the questionnaire, were explained to the participants if necessary. The completeness of each questionnaire was verified by the researcher daily.

### 2.8. Data Analysis

Data were analyzed using the IBM Statistical Package for Social Sciences (SPSS) version 23.0. Descriptive statistics were compiled to calculate percentages, frequencies, ranges, and means of the quantitative variables. Bivariate analysis was performed, and only variables with a *p*-value < 0.05 were declared statistically significant. Those that met statistical significance were included in the multiple logistic regression analysis models to compute the adjusted Odds Ratio (aOR) at 95% CI (Confidence Interval). To be specific, Pearson product moment correlation and Chi square analysis were used to estimate relationships or associations across selected study variables. An independent sample t-test was used to determine mean differences.

### 2.9. Ethical Issue

The researcher obtained approval from the Department of Microbiology at the Xiangya School of Public Health, Central South University Committee at the School of Medicine and Allied Health Sciences, University of The Gambia. Informed consent was given to each participant to sign or thumbprint depending on their preferences prior to recruitment into the study. The decision to sign the consent form is made by each participant without the researcher’s interference. Participants’ privacy and confidentiality were maintained throughout the study. Each of the study participants had the right to withdraw from the study at any point in time during the course of the study without their rights being infringed/affected.

## 3. Results

### 3.1. Socio-Demographic Characteristics of the Participants

A total of 339 women participated in the study, in which more than half (53.7%) were from the western region. Kanifing Municipality Council accounts for 16.5% of the total participants followed by the North Bank Region (12.4%) as shown in Appendix A. Furthermore, 98.8% of the last-born children of the participants were alive, in which the majority were male (51%). Regarding the marital status of participants, 92.3% of the respondents were married and 4.7% were single. With regards to occupation, the majority (72.3%) of the respondents (mothers) were housewives, and the occupation of fathers accounted for 62.5%. Furthermore, 78.2% of the participants resided in the municipality regulatory area and 12.1% lived in the moderate municipality regulatory area. Regarding settlement type, the majority (92%) lived in urban settlements. The majority of the respondents (56.6%) have an in-house bathroom and 42.2% had a bathroom a short distance behind the house (Appendix A).

The mean age of the respondents (m”ther’) was 29 years, fathers were 39 years, and the average length of marriage was 9 years. The average age of the last child was 4 years. The average monthly income of the household was D6161.7 (equivalent to $100) ± 27,797.8. Furthermore, the average number of people living in a household was 6, and the number of children under-five was 1.

### 3.2. Prevalence of Key Outcome Variables

Please provide a simple cluster bar chart for the core outcomes variables of the paper as initially recommended by reviewer 1.

### 3.3. Psychosocial Skills of Offspring Aged 3 to 5 Years

The majority of the respondents’ infants had a good health profile or record. For example, only 11.8% of the respondents’ infants were malnourished, 4.4% had low hemoglobin, 0.3% had sickle-cell disease, and 4.4% had asthma.

Neonatal complications can pose serious threats to the survival of the infant, especially in the first 100 days. Our findings showed that 10.6% of the infants had poor suckling abilities, 15.9% had a fever, 5.6% had breathing problems, and 8.96% had low birth weight.

Frequency analysis showed that 39.2% of respondents’ offspring had very good social skills and 33% had excellent social skills. Regarding communication skills, 37.6% of the respondents’ offspring had very good skills followed by 30.6% with good skills on average. Moreover, 35.3% had good cognitive skills followed by 28.8% with excellent cognitive skills. With regards to motor skills, 31.8% had excellent skills. Regarding activities of daily living, 38.3% had excellent skills (Table 1).

### 3.4. Health Profile of Mothers’ Preconception and Conception

The health profile of mothers in intrapartum and postpartum periods showed that 79.1% had normal labor, 90.6% had a normal delivery, 77.3% had no excessive hemorrhage, and 7.4% also experienced eclampsia. The majority (66.4%) of the respondents practiced exclusive breastfeeding (see Table 1 for more detail).

The majority complained of early morning vomiting (66.4%) and breast engorgement, 28.6% had a history of low hemoglobin, and 11.2% had chronic high blood pressure. History of STI was observed for 8.3% of the respondents (Table 2).

### 3.5. Smoking Habit as Predictors to the Psychosocial Skills of Offspring

According to respondents, 21.5% of their spouses had smoked before and only 6.5% worked in an industrial area. Furthermore, 18.3% of the male counterparts were active smokers, and 9.7% of them smoke at home.

To determine the association between smoking habits or socio-demographic variables and the psychosocial skills of offspring, Chi-square and binary logistics regression analyses were conducted. The chi-square results showed that the father’s history of smoking was not associated with the level of psychosocial skills of offspring (X^2^ = 1.893, *p* = 0.169). Furthermore, the correlation coefficient was not significant (r = 0.075, *p* = 0.170). However, the occupation of fathers (working in an industrial area) was significantly associated with the psychosocial skills of the offspring (X^2^ =11.928, *p* = 0.001). There was a significant weak correlation coefficient between occupations and psychosocial skills of offspring (r = 0.188, *p* = 0.001). The other variables, as shown in Table 3, are not significantly associated with the psychosocial skills of offspring.

The binary logistic regression also showed no association between the smoking habit of the parent (father) and the psychosocial skills of their offspring. However, parents or fathers who worked in industrial areas were more likely to have offspring with poor psychosocial skills [OR = 5.763 (95% CI; 1.907, 17.414), *p* = 0.002] (Table 3).

### 3.6. Chemical Exposure as Predictors to Psychosocial Skills of Offspring

Exposure to chemicals by parents, especially the mother, can predispose the infant or fetus to medical complications. In fact, 60.5% of our respondents were exposed to insecticides, DDT, mosquito spray, or coil, 70.2% to heterocyclicamines, and 79.9% to Aflatoxin. Furthermore, approximately 24.5% of the respondents reside by the roadside.

To ascertain the association between exposure to chemicals and the state of psychosocial skills of offspring, a Chi-square test and logistics regression analysis were performed. The chi-square analysis showed that oxidative stress was significantly associated with the psychosocial skills of offspring (X^2^ =7.251, *p* = 0.007). There was a statistically significant weak and negative correlation (r= −0.146) between oxidative stress and the psychosocial skills of offspring.

Table 4 shows the logistic regression used to predict factors of chemical exposure and their association with the psychosocial skills of the offspring. The analysis revealed that mothers who are exposed to oxidative stress were more likely to have offspring with poor psychosocial skills [OR = 0.411(95% CI, 0.212, 0.796), *p* = 0.008]. Furthermore, mothers exposed to high temperatures were more likely to have offspring with poor psychosocial skills [OR = 0.43 (95% CI; 0.273, 0.679), *p* = 0.001].

### 3.7. Correlation between Maternal Stress and Psychosocial Skills of Infants

The majority (53.1%) of the respondents had no stress before pregnancy. Among the stressors, financial worries such as food, shelter, and transportation accounted for 36.9% cumulatively, compared to other stressors, followed by sleeping difficulties and other psychosomatic symptoms (26.8%). The least stressful were problems related to their current or previous pregnancy, in which 92.9% of respondents indicated no stress, followed by “having to move or change of settlements, either recently or in the future” (91.7%).

As to the level of stress among respondents during pregnancy, the analysis showed that 44.2% of the participants had stress during pregnancy, which is lower compared to before pregnancy. Sleeping difficulties and other psychosomatic symptoms and financial worries such as food, shelter, and transportation are the major stressors during pregnancy. To determine the correlation between the psychosocial skills of offspring and maternal stress before and during pregnancy, a bivariate analysis was performed. The result showed a statistically significant weak correlation between maternal stress before pregnancy and the psychosocial skills of offspring (r= −142, *p* = 0.009). Furthermore, there was a negative correlation between maternal stress during pregnancy and the psychosocial skills of offspring (r= −147, *p* = 0.007) (Table 5).

### 3.8. Nutritional Status of Mothers against Psychosocial Skills of Offspring

Supplements and vaccines are very important in the prevention and control of maternal diseases and conditions. In this study, the majority of the participants received tetanus toxoid (69%), 65.5% received iron supplementation, and 71.4% received fansidar as a preventive therapy. However, the majority of the respondents did not receive zinc supplementation (65.5%) and antibiotic prophylaxis (54.6%). More than half (51.9%) of the respondents acknowledged the use of excess nutrients before or during pregnancy. Moreover, 66.4% partake in a low-protein diet, while 84.4% also have a high-fat diet.

The binary logistics regression analysis was conducted to determine possible predictors of the outcome variable (psychosocial skills of the offspring). Mothers who consumed a low-protein diet were almost three times more likely to have infants with good psychosocial skills (OR = 2.545, 95% CI: 1.529–4.237) (Table 6).

## 4. Discussion

Globally, in addition to adult obesity, childhood obesity has reached epidemic proportions. This trend not only affects physical health but also has far-reaching and economic implications. While some healthy lifestyle interventions can mitigate these serious metabolic disorders, there seems to be a race between these interventions and the passing down of these deleterious traits to the next generation through epigenetic changes. Furthermore, epigenetic changes have similar consequences as intergenerational and transgenerational changes in the brain function of the offspring. While much attention is paid to maternal epigenetic effects during gestation and paternal effects [20]. What lies behind the illusion is biological processes with negative effects of trauma across generations, thereby identifying risk groups in intergenerational transmission of mental disorders and sorts [21]. Elucidating these occurrences in the population base will provide a clue to the biological mechanistic pathways as useful tools for their prevention [22,23].

Neonatal complications can pose serious threats to the survival of the infant, especially in the first 100 days. Our findings showed that a good number of the infants had poor suckling, fever, breathing problems, and low birth weight. In addition to the episodes of infectious diseases common during the rainy season, the most extended endurance of eating the previous harvest when food stocks are depleted avails a chronically marginal diet. Previous study conducted in Gambia showed an increased death rate of individuals born in nutritionally poor seasons and this has been related to infections [3]. Reproductive women tend to lose weight (3–6 kg) due to the burden of laborious seasonal farm work. These seasonal average weights for pregnant women are better during the harvest season [24,25]. This reduced weight leads to lower birth weight in infants born during the hunger season as compared to those born during harvest seasons [26,27,28]. From this study, we found that the majority of the infants had good health records. However, a considerable number had low hemoglobin and asthma, and several had sickle-cell disease. Despite the fact that most of the respondents’ offspring had good social skills, some were found to fall under good social skills, and almost the same proportions hold for communication and cognitive skills. Furthermore, a good number of offspring were found with good motor skills and activities of daily living. Exposure to chemicals by parents, especially the mother, can predispose the infant or fetus to some medical complications. More than half of our study respondents were exposed to insecticides, DDT, mosquito spray, or coil and an even greater number were found to be exposed to heterocyclicamines and aflatoxin during the course of pregnancy. Cypermethrin insecticide exposure was found to be correlated with long-lasting reproductive malfunctions in female mice generations. Up to the F2 generation of the female line, developmental abnormalities were observed, and this calls for a divergence from the molecular mechanism studies on trans-generational sex-linked [29]. In addition, many reside by the roadside, where they will be exposed to smoke coming from exhaust engines. Even a low dose of lead or Polybrominated diphenyl ethers/lead mixture exposure was seen with behaviour repetitive patterns, and learning challenges in male mice. Moreover, the systemic inflammatory response was also synergistic [30]. From our study, there was no association between the smoking habits of parents and the psychosocial skills of their offspring. However, parents or fathers who worked in industrial areas were more likely to have offspring with poor psychosocial skills.

Although the pathophysiological pathway that could shed some light on the association between maternal stressors and offspring’s neuropsychiatric illness is not fully elucidated, there is evidence from epidemiological and animal models in addition to infection and maternal overnutrition. Maternal psycho psychiatric stress puts the offspring at increased neuropsychiatric risk [7]. At least more than half of the respondents had no stress before pregnancy. However, among the stressors, financial worries such as food, shelter, and transportation accounted for one-third, followed by sleeping difficulties and other psychosomatic symptoms. The least stressful were problems related to the current or previous pregnancy in which almost all the respondents indicated no stress, while “having to move or change of settlements, either recently or in the future” was also not a big problem among the respondents. It is obvious that with this kind of prenatal maternal stress, exposure timing could determine the infant’s immune epigenetic profiles [31]. There was a statistical significance between paternal early life stress (ELS) and the development of offspring’s brain development. Thus, there is a need to expand pediatrics studies on the intergenerational inheritance of ELS in generations related to brain development [32]. Prenatal stress alters metabolic pathways in protein and energy metabolism, and these are predictive of an increased risk of insulin resistance, obesity, and diabetes [33]. A good number of our study respondents had increased stress during pregnancy. These could be caused by sleeping difficulties and other psychosomatic symptoms. In addition, financial worries such as food, shelter, and transportation were major stressors during pregnancy. From our study findings, we could detect a negative correlation between maternal stress before pregnancy or during pregnancy and the psychosocial skills of offspring. In contrast to our study findings, at 16 months after birth, the infants were seen with lower fine motor development and difficult temperament. However, the same study could not find any association between infant salivary cortisol and prenatal maternal distress [34]. In predicting the factors associated with psychosocial skills from our study, mothers who are exposed to oxidative stress and high temperatures are more likely to have offspring with poor psychosocial skills.

Supplements and vaccines are very important in the prevention and control of maternal diseases and conditions. Acetaminophen or paracetamol, as one of the common medicines taken during pregnancy, has an epidemiological background association with risk for neurodevelopmental disorders, asthma, genital malfunctions, and behavioral changes in offspring [35]. In this study, the majority of the participants received tetanus toxoid, iron supplementation, and fansidar. However, the majority of the respondents did not receive zinc supplementation and antibiotic prophylaxis. Moreover, maternal zinc status did not influence developmental outcomes in children as previously thought with central nervous system development [22]. Similar to other middle-income countries, there are seasonal nutritional fluctuations. This substantial phenomenon is experienced naturally in some rural farming communities. Studies found this to be causing maternal undernutrition, especially during prolonged dry periods, locally called the ‘hungry season’ in Gambia [17]. Eating disorders are associated with an increased risk of adverse outcomes such as preterm delivery, miscarriage, poor fetal growth, or malformations [36]. Immune dysfunction in autistic offspring via superoxide dismutase 2 suppression is found to be induced by maternal diabetes [37]. Maternal diabetes-induced neurodevelopmental disorders such as autism-related phenotypes in offspring are potentiated by vitamin D deficiency [38]. In this study, more than half of the respondents acknowledged the use of excess nutrients before or during pregnancy, a high-protein diet, and a high-fat diet. Mothers who consumed a high-protein diet were almost three times more likely to have infants with good psychosocial skills.

Although the study involved the mothers by going through the antenatal card to read through the ANC while ascertaining some medical history, the level of literacy and health education among mothers in Gambia is relatively low. This has serious implications on the credibility of some information and how representative the sample size might be. Exploring the intergenerational effects of gestational nutrient restriction and its effect on the population will require decades to conduct a follow-up study. Furthermore, ensuring a reliable disease history requires data, which are very limited. In response to this, a follow-up or retrospective study becomes very necessary. While this study is an observational study, there was not enough funds to enable an increased sample size to overcome the biases in sampling and responses. Epidemiological evidence, instead of genetic data, should be added to the scientific literature. Despite the fact that well-experienced community nurses break down questionnaires through home visit interviews, due to the high illiteracy rate, some participants were not able to recall or identify specific cardiometabolic diseases.

This emerging paradigm shift in the science of epigenetic modifications has a permanent imprint on phenotypic responses [39]. It is obvious that nutrition, stress status, and parental metabolic conditions altered epigenetic intergenerational transmission. While this is trending, we strongly recommend policy implications that will reduce the continuous disadvantages across generations. This will no doubt check the insights into the perpetuation of compromised lives across generations. With a larger sample size study, a well-designed structured intervention could be established during each stage of pregnancy, and also during the postpartum period. While addressing diets seem more feasible compared to developing new pharmacological targets, dietary monitoring connects physiological pathways in the complex human body with positive effects and can help metabolic, as well as mental, disorders. It is important to include primary lifestyle changes among reproductive-age women, which should focus on dietary selections and switching dietary options during nutrition restrictions times. Furthermore, there should be an increased level of awareness among mothers about their offspring’s health and their husbands’ lifestyles, which might influence the adulthood health of offspring.

### Limitation of the Study

The key limitation of this study was that the study shows association but not causality that could be limited in distinctively explaining relationship between variables of interest in the study. There are some possibilities of recalled bias regarding their history of exposure.

## 5. Conclusions

The developmental origin of the health of young adults is based on socioeconomic disadvantages and maternal prenatal distress. This affects both the short- and long-term mental and physical health of the offspring. However, the biological system involved in the transmission of these factors across generations is not fully documented due to the utero timeframe. The epigenetic pathways of parental childhood experiences of these phenomena are only beginning to be studied in humans, and translational research with a viewpoint on human cohort studies based on solid study design and valid methodological approaches combined with the existing longitudinal studies should be considered. This study provided evidence-based findings on the influence of parental energy, toxicant exposure, age, and nutrient restriction on offspring growth in Gambia. The study was able to ascertain if the maternal diet during gestation, toxicant exposure, maternal stress, and parental smoking habits influence the early life of offspring.

## Figures and Tables

**Table 1 ijerph-19-13045-t001:** Psychosocial skills of offspring between 3 to 5 years (n = 339).

Psychosocial Skills		Poor (%)	Good (%)	Very Good (%)	Excellent (%)
Social skills					
	Eye contact	3 (0.9)	91 (26.8)	141 (41.6)	104 (30.7)
	Joint attention	4 (1.2)	102 (30.1)	136 (40.1)	97 (28.6)
	Responding to adult direction	2 (0.6)	69 (20.4)	142 (41.9)	126 (37.2)
	Recognize emotional states	3 (0.9)	137 (40.4)	84 (24.8)	115 (33.9)
	Peer friendships	0 (0.0)	61 (18.0)	161 (47.5)	117 (34.5)
	Average scores	0.7	27.1	39.2	33
Communication skills					
	Shift gaze of person to object	20 (5.9)	127 (37.5)	117 (34.5)	75 (22.1)
	Gesturing	2 (0.6)	102 (30.1)	154 (45.4)	81 (23.9)
	Pointing	2 (0.6)	75 (22.1)	131 (38.6)	131 (38.6)
	Expression of emotions	5 (1.5)	111 (32.7)	108 (31.9)	115 (33.9)
	Average scores	2.2	30.6	37.6	29.6
Cognitive skills					
	Readiness skills	8 (2.4)	135 (39.8)	111 (32.7)	85 (25.1)
	Object permanence or recalling	29 (8.6)	106 (31.3)	101 (29.8)	103 (30.4)
	Smartness (Concept development)	28 (8.3)	92 (27.1)	107 (31.6)	112 (33.0)
	Decision making	45 (13.3)	145 (42.8)	58 (17.1)	91 (26.8)
	Average scores	8.2	35.3	27.6	28.8
Motor skills					
	Gross motor	29 (8.6)	129 (38.1)	77 (22.7)	104 (30.7)
	Playing on playground equipment	0 (0.0)	61 (18.0)	137 (40.4)	141 (41.6)
	Jumping	10 (2.9)	59 (17.4)	166 (49.0)	104 (30.7)
	Ball catching	69 (20.4)	108 (31.9)	79 (23.3)	83 (24.5)
	Walking up/downstairs	31 (9.1)	144 (42.5)	58 (17.1)	106 (31.3)
		8.2	29.6	30.5	31.8
Activities of daily living					
	Independent feeding	4 (1.2)	114 (33.6)	67 (19.8)	154 (45.4)
	Toilet training	36 (10.6)	69 (20.4)	99 (29.2)	135 (39.8)
	Clothes on/off independently	22 (6.5)	74 (21.8)	124 (36.6)	119 (35.1)
	Hand-washing abilities	36 (10.6)	123 (36.3)	69 (20.4)	111 (32.7)
		7.2	28	26.5	38.3

Note: Figures in brackets are in percentages.

**Table 2 ijerph-19-13045-t002:** The health profile of the mother during preconception and conception (including obstetric and medical histories).

Disease/Conditions	No (%)	Yes (%)
Was there a miscarriage?	292 (86.1)	47 (13.9)
Early morning vomiting	114 (33.6)	225 (66.4)
Breast engorgement	275 (81.1)	64 (18.9)
Missed Period	82 (24.2)	257 (75.8)
Edema	317 (93.5)	22 (6.5)
Chronic high blood pressure	301 (88.8)	38 (11.2)
History of low hemoglobin (anemia)	242 (71.4)	97 (28.6)
History of sickle cell disease	331 (97.6)	8 (2.4)
History of STI	311 (91.7)	28 (8.3)
Diabetes mellitus	333 (98.2)	6 (1.8)
Diabetes mellitus (father)	330 (97.3)	9 (2.7)
Lung cancer (mother)	339 (100.0)	0 (0.0)
Peptic ulcer diseases	312 (92.0)	27 (8.0)
Paternal obesity	301 (88.8)	38 (11.2)
Lung cancer (father)	336 (99.1)	3 (0.9)
Asthma	320 (94.4)	19 (5.6)
Tuberculosis	338 (99.7)	1 (0.3)

**Table 3 ijerph-19-13045-t003:** Smoking habits as predictors of psychosocial skills of offspring.

Variable	n (%)	Unadjusted OR * (95% CI)	*p*-Value	Adjusted OR (95% CI)	*p*-Value
Father ever smoke					
	No (307)		Ref		
	Yes (32)	1.439 (0.856, 2.419)	0.117	0.91 (0.36, 2.303)	0.842
Father working at industrial area (pollution)					
	No (317)		Ref		
	Yes (22)	5.763 (1.907,17.414)	0.002 *	4.832 (1.5, 15.563)	0.008 *
Active smoking parent					
	No (277)		Ref		
	Yes (62)	1.52 (0.874, 2.644)	0.138	0.981 (0.342, 2.816)	0.972
Parent smokes in the house					
	No (306)		Ref		
	Yes (33)	1.898 (0.911, 3.953)	0.087	1.246 (0.339, 4.577)	0.741
Parent smokes around you while you are pregnant					
	No (306)		Ref		
	Yes (33)	1.898 (0.911, 3.953)	0.087	1.231 (0.368,4.123)	0.736
Smokes before and now quit					
	No (266)		Ref		
	Yes (73)	2.068 (0.977, 4.379)	0.058	1.574 (0.656, 3.773)	0.309

OR = Odds Ratio and CI = confidence interval, * Statistical significance, coding 0 = No and 1 = yes, 0 = poor psychosocial skills and 1 = good psychosocial skills.

**Table 4 ijerph-19-13045-t004:** Exposure to chemicals and psychosocial skills of offspring.

Variable	Category	Psychosocial Skills		Unadjusted OR (95% CI)	*p*-Value	Adjusted OR (95% CI)	*p*-Value
		Poor (n)	Good (n)				
Insecticides, DDT, mosquito spray or mosquito coil							
	No	65	69	Ref			
	Yes	117	88	0.709 (0.458, 1.097)	0.123	0.707 (0.442, 1.13)	0.147
Heterocyclicamines (2-cooked meat products)							
	No	46	55	Ref			
	Yes	136	102	0.627 (0.393, 1.002)	0.051	0.871 (0.515, 1.471)	0.604
Aflatoxin (eating too much peanut soup)							
	No	37	31	Ref			
	Yes	182	157	1.037 (0.608, 1.768)	0.893	1.507 (0.845, 2.686)	0.165
Oxidative stress							
	No	147	143	Ref			
	Yes	35	14	0.411 (0.212, 0.796)	0.008 *	0.521 (0.282, 1.06)	0.072
High temperature							
	No	97	114	Ref			
	Yes	182	157	0.43 (0.273, 0.679)	0.001 *	0.47 (0.282, 0.782)	0.004 *
Residing at the roadside during pregnancy							
	No	131	125	Ref			
	Yes	51	32	0.658 (0.397, 1.09)	0.104	0.933 (0.536, 1.625)	0.808

OR = Odds Ratio and CI = confidence interval, * Statistical significance Coding: 0 = No, 1 = Yes.

**Table 5 ijerph-19-13045-t005:** Correlation between maternal stress and psychosocial skills of offspring (n = 339).

		Overall Psychosocial Profile of Offspring	Maternal Stress before Pregnancy	Maternal Stress during Pregnancy
Overall psychosocial profile of offspring	r			
	*p*-value			
Maternal stress before pregnancy	r	−0.142 **		
	*p*-value	0.009		
Maternal stress during pregnancy	r	−0.147 **	0.755 **	
	*p*-value	0.007	0.001	

** statistical significance.

**Table 6 ijerph-19-13045-t006:** Nutritional status as predictors of psychosocial skills of offspring (n = 339).

Question	B	Wald	Sig.	OR	95% C.I	
					Lower	Upper
Excess nutrient intake	−0.084	0.129	0.72	0.92	0.583	1.452
High-protein diet (meat, fish, milk)	0.934	12.911	0.001 *	2.545	1.529	4.237
High-fat diet (butter, fries, cookies)	0.18	0.291	0.589	1.197	0.622	2.304
Vegetable oil	0.209	0.301	0.583	1.233	0.584	2.602
Red meat	0.718	3.877	0.049 *	2.05	1.003	4.19
Dietary fiber (raw fruits, vegetables)	0.138	0.04	0.841	1.148	0.300	4.393
Rich vitamin foods (fruits, vegetables)	0.812	1.678	0.195	2.252	0.659	7.688
Taking supplements (e.g., vitamin)	0.106	0.185	0.667	1.112	0.686	1.801
Fasting	0.288	1.089	0.297	1.333	0.777	2.288

OR = Odds Ratio and CI = confidence interval, * Statistical significance.

## Data Availability

Not applicable.

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
