# Peer review of "The Influence of Parental Environmental Exposure and Nutrient Restriction on the Early Life of Offspring Growth in Gambia—A Pilot Study"

_ijerph, 2022, doi:10.3390/ijerph192013045_

Round 1
Reviewer 1 Report
The article presents interesting results, but it has to organize better the methodology as well as to have more consistency in presenting the results.
The following changes are needed:
1. Abstract
The methodology should describe briefly what kind of study/sample/instrument was used
The abstract has to be self explanatory, now it is not celar what it was done
2. Methodology
The instrument for dat collection should be presented, now it is not celar what kind of issues were investigated and how/ which questions/what posibilities of answers
The scales and ways of classification ( malnutrition, level of haemoglobin, several skills, stress, etc) should be presented , now it is not celar what methodology/cut-off points were used for data interpretation
Data analyses should be more clear presented and I suggest to have consistency in how data are analised, now there are issues which are analysed with bivariate correlations, other with logistic regression, while in the methodology it is mentioned that both are made and in the logistic regression are included only variables which are statistical significant at bivariate corelation. I sugest to describe first the univariate analyses, then the bivariate corelations ( including what type/coeficients, how the results are interpreted) and then the logistic regression ( what kind of dependent and independent variables, how the variables were included in the analyses)
3. Results
More consistency and clarity is needed, for each issue should be presented the prevalences, then the bivariate correlation and then regresion analyses. Tables should specify below them the coding used for variables, to be able to understand them
Author Response
Reviewer 1
Abstract
The methodology should describe briefly what kind of study/sample/instrument was used
Response: The revised version of the manuscript has in the abstract; the study/sample/instrument used for your perusals.
The abstract has to be self explanatory, now it is not celar what it was done
Response: The entire abstract is revised for a better self explanatory
Methodology
The instrument for dat collection should be presented, now it is not celar what kind of issues were investigated and how/ which questions/what posibilities of answers
Response: This study involved structured interview questionnaire, which was developed by the research team. These team includes public health professionals, nurses and medical students of the University of Gambia and they conducted three months long interview by house visit while assessing the antenatal care (ANC) cards to supplement some information.
The scales and ways of classification (malnutrition, level of haemoglobin, several skills, stress, etc) should be presented, now it is not celar what methodology/cut-off points were used for data interpretation
Response: Since it is a qualitative study and not much scales are available as per data given, we measure the variables as low and high. In our subsequent study, we hope introduce score sheet to determine the cut-off points.
Data analyses should be more clear presented and I suggest to have consistency in how data are analised, now there are issues which are analysed with bivariate correlations, other with logistic regression, while in the methodology it is mentioned that both are made and in the logistic regression are included only variables which are statistical significant at bivariate corelation. I sugest to describe first the univariate analyses, then the bivariate corelations (including what type/coeficients, how the results are interpreted) and then the logistic regression (what kind of dependent and independent variables, how the variables were included in the analyses)
Response: These are adjusted and supplemented in the revised manuscript
Results
More consistency and clarity is needed, for each issue should be presented the prevalences, then the bivariate correlation and then regresion analyses. Tables should specify below them the coding used for variables, to be able to understand them
Response: The dependent variable for the study is psychosocial skills of offspring, which was categorized into Good or poor psychosocial skills. The independent variables are the socio-demographic variables such as smoking habits and exposure to chemicals. Chi square analysis was done to determine association between these categorical variable and also determine the correlation coefficient. Binary logistics regression was also done to identify predictors of psychosocial skills of offspring
Reviewer 2 Report
Banjika et al., 2022 describes the possible of effect of the external environmental factors affecting the early lives of offsprings. They conducted a questionnaire where about 339 participants were enrolled.
The methods employed have not been fully described. Of importance, the experimental design needs improvement. A flowchart describing the study design will help immensely.
Creating this flow chart will assist in identifying biases or factors which may have acted in combination and whether these were removed. Have associations been ruled out, for instance did you have a cohort that neither smoked but worked in industry and vice versa? In that same breath, the article does not specify if, for example, smoking and working in an industrial site played a role. It does not look at where some of these factors intersected as they possibly might have. The methodology described is purely descriptive but there are factors which can confound the research outputs. The issue of infectious disease is only referenced to, in the discussion which should have been looked at or described fully in the methods, and maybe included as an exclusion criteria or at best a limitation of the study. However, it is discussed in the section - discussion.
On the discussion the point above was shown in the first few lines of the discussion as well as in opening statements in the different paragraphs. The essence of the discussion is to present your findings and discuss those in relation to previous literature. For example on toxicant exposure, the authors describe studies done in mice and then talk of their results. Also, the results were not supporting this finding and that is where the discussion becomes important. The absence of line numbers make it very difficult to refer to exact sections in the manuscript. The point on parents wrking in industries was not discussed at all. Similarly, the negative correlation between maternal stress during pregnancy and psychosocial skills of offspring was not fully discussed.
My general recommendation is the study design needs improvement. The sample size must be greatly increased to facilitate for this. Confounders should be removed or included - like presence of infectious diseases. If associations have been made from previous papers, considering using Bayesian statistics may be helpful. This study holds a great potential and warrants the investment in improving design and increasing the sample size.
Author Response
Reviewer 2
Banjika et al., 2022 describes the possible of effect of the external environmental factors affecting the early lives of offsprings. They conducted a questionnaire where about 339 participants were enrolled.
The methods employed have not been fully described. Of importance, the experimental design needs improvement. A flowchart describing the study design will help immensely. Creating this flow chart will assist in identifying biases or factors which may have acted in combination and whether these were removed. Have associations been ruled out, for instance did you have a cohort that neither smoked but worked in industry and vice versa? In that same breath, the article does not specify if, for example, smoking and working in an industrial site played a role. It does not look at where some of these factors intersected as they possibly might have.
Response: A flow chart is included with respect to the experimental design
The methodology described is purely descriptive but there are factors which can confound the research outputs.
Response: A more detailed methodology is supplemented in the revised manuscript
The issue of infectious disease is only referenced to, in the discussion which should have been looked at or described fully in the methods, and maybe included as an exclusion criteria or at best a limitation of the study. However, it is discussed in the section - discussion.
Response: For the fact the participants are mostly women with no clear conscience as to health and diseases, and most lost the ANC cards that would have specify the infectious diseases associated, the discussion section made reference to a similar study conducted in the same country. ‘Periconceptional maternal micronutrient supplementation is associated with widespread gender related changes in the epigenome: a study of a unique resource in the Gambia.’
On the discussion the point above was shown in the first few lines of the discussion as well as in opening statements in the different paragraphs. The essence of the discussion is to present your findings and discuss those in relation to previous literature. For example on toxicant exposure, the authors describe studies done in mice and then talk of their results. Also, the results were not supporting this finding and that is where the discussion becomes important. The absence of line numbers make it very difficult to refer to exact sections in the manuscript. The point on parent’s wrking in industries was not discussed at all. Similarly, the negative correlation between maternal stress during pregnancy and psychosocial skills of offspring was not fully discussed.
Response: The entire discussion is re-written and the above concerns are solved including line numbering.
My general recommendation is the study design needs improvement. The sample size must be greatly increased to facilitate for this. Confounders should be removed or included - like presence of infectious diseases. If associations have been made from previous papers, considering using Bayesian statistics may be helpful. This study holds a great potential and warrants the investment in improving design and increasing the sample size.
Response: While this is the pilot study, a more extensive study is design is being developed that includes a robust specific environmental factors.

Round 2
Reviewer 1 Report
The authors have tried to improve this version, but many comments and sugesstions remained without a satisfactory change/clarification
-The instrument for dat collection should be presented, now it is not clear what kind of issues were investigated and how/ which questions/what posibilities of answers
The authors say something about interviews and cards, but the the questions/investigated issues should be clearly described, please look in other articles published in the journal for several examples about how to organise this type of information
-The scales and ways of classification (malnutrition, level of haemoglobin, several skills, stress, etc) should be presented, now it is not celar what methodology/cut-off points were used for data interpretation
The authors answered it is a qualitative study, this answer is not acceptable since they have statistics involved they used scales and numbers, these aproaches should be explained ( what is low/high, who decided this-the researchers, the participants?)
.-Data analyses should be more clear presented and I suggest to have consistency in how data are analised, now there are issues which are analysed with bivariate correlations, other with logistic regression, while in the methodology it is mentioned that both are made and in the logistic regression are included only variables which are statistical significant at bivariate corelation. I sugest to describe first the univariate analyses, then the bivariate corelations (including what type/coeficients, how the results are interpreted) and then the logistic regression (what kind of dependent and independent variables, how the variables were included in the analyses)
-More consistency and clarity is needed, for each issue should be presented the prevalences, then the bivariate correlation and then regresion analyses. Tables should specify below them the coding used for variables, to be able to understand them
Author Response
Response to Reviewers’ Comments
Reviewer 1
Comments: The instrument for data collection should be presented, now it is not clear what kind of issues were investigated and how/ which questions/what possibilities of answers
Response: We have generously expanded the scope of our data collection description on the types of information elicited using the said tool as requested.
Comments: The authors say something about interviews and cards, but the questions/investigated issues should be clearly described, please look in other articles published in the journal for several examples about how to organize this type of information
Response: This section is revised and we’ve added relevant information about the two data collection tools (Structured interview questionnaire and IWC card) used for this study. Each of them were sufficiently described in their separate sub-sections under data collection tools section of the manuscript. Thank you.
Comments:The scales and ways of classification (malnutrition, level of haemoglobin, several skills, stress, etc) should be presented, now it is not clear what methodology/cut-off points were used for data interpretation
Response: Thank you for that insightful comment. We now added a new section called study variables which is further dichotomized into outcome and explanatory variables. Each of the outcomes variables including their reference/cut-off points were explicitly described as expected.
Comments:The authors answered it is a qualitative study, this answer is not acceptable since they have statistics involved they used scales and numbers, these approaches should be explained (what is low/high, who decided this-the researchers, the participants?)
Response: We have worked on this as this a population-based analytical cross-sectional study design and we used the referenced/cut-off points to term our decision line including the conversion of computed cumulative scores as proportion which were expressed in percentages and classified accordingly as per reference points.
Comments: Data analyses should be more clear presented and I suggest to have consistency in how data are analised, now there are issues which are analysed with bivariate correlations, other with logistic regression, while in the methodology it is mentioned that both are made and in the logistic regression are included only variables which are statistical significant at bivariate corelation. I sugest to describe first the univariate analyses, then the bivariate corelations (including what type/coeficients, how the results are interpreted) and then the logistic regression (what kind of dependent and independent variables, how the variables were included in the analyses)
Response: Thank you for the insightful comments. We have now re-arranged the sequencing of our results section as per the required statistical standards. We first present our univariate analysis, prevalence summaries, bivariate analysis using chi-square/fisher exact test then correlation and regression analysis depending on meeting their preconditions/assumptions. This makes it easy to follow and understand the results with ease.
Comments: More consistency and clarity is needed, for each issue should be presented the prevalence, then the bivariate correlation and then regression analyses. Tables should specify below them the coding used for variables to be able to understand them
Reference: Currently outlined in the manuscript for areas that need to be adjusted. Once all those are being addressed, then we can composed a befitting response to this point.
Reviewer 1
Comments: The instrument for data collection should be presented, now it is not clear what kind of issues were investigated and how/ which questions/what possibilities of answers
Response: We have generously expanded the scope of our data collection description on the types of information elicited using the said tool as requested.
Comments: The authors say something about interviews and cards, but the questions/investigated issues should be clearly described, please look in other articles published in the journal for several examples about how to organize this type of information
Response: This section is revised and we’ve added relevant information about the two data collection tools (Structured interview questionnaire and IWC card) used for this study. Each of them were sufficiently described in their separate sub-sections under data collection tools section of the manuscript. Thank you.
Comments:The scales and ways of classification (malnutrition, level of haemoglobin, several skills, stress, etc) should be presented, now it is not clear what methodology/cut-off points were used for data interpretation
Response: Thank you for that insightful comment. We now added a new section called study variables which is further dichotomized into outcome and explanatory variables. Each of the outcomes variables including their reference/cut-off points were explicitly described as expected.
Comments:The authors answered it is a qualitative study, this answer is not acceptable since they have statistics involved they used scales and numbers, these approaches should be explained (what is low/high, who decided this-the researchers, the participants?)
Response: We have worked on this as this a population-based analytical cross-sectional study design and we used the referenced/cut-off points to term our decision line including the conversion of computed cumulative scores as proportion which were expressed in percentages and classified accordingly as per reference points.
Comments: Data analyses should be more clear presented and I suggest to have consistency in how data are analised, now there are issues which are analysed with bivariate correlations, other with logistic regression, while in the methodology it is mentioned that both are made and in the logistic regression are included only variables which are statistical significant at bivariate corelation. I sugest to describe first the univariate analyses, then the bivariate corelations (including what type/coeficients, how the results are interpreted) and then the logistic regression (what kind of dependent and independent variables, how the variables were included in the analyses)
Response: Thank you for the insightful comments. We have now re-arranged the sequencing of our results section as per the required statistical standards. We first present our univariate analysis, prevalence summaries, bivariate analysis using chi-square/fisher exact test then correlation and regression analysis depending on meeting their preconditions/assumptions. This makes it easy to follow and understand the results with ease.
Comments: More consistency and clarity is needed, for each issue should be presented the prevalence, then the bivariate correlation and then regression analyses. Tables should specify below them the coding used for variables to be able to understand them
Reference: Currently outlined in the manuscript for areas that need to be adjusted. Once all those are being addressed, then we can composed a befitting response to this point.
Reviewer 2 Report
Thank you authors for attempting to answer all my questions.
There are still changes required in the manuscript. To start with the study design as depicted by the appended flowchart misses the point for which I suggested why we needed the flowchart. The reason is, and I will make an example, did you have a cohort that neither smoked but worked in industries? The study must be clear. Even though smoking played no role (as per p value), did you do associations as well? For example smoking and working in industry. Here, I present two situations that can confound the findings.
Again, the fact that Antenatal cards could not be found for some participants, presents a limitation. This limitation really throws some of the findings into dispute. I refer to the contribution of infectious diseases.
If the study is a pilot study, then the title should reflect that.
Author Response
Response to Reviewers’ Comments
Reviewer 2
Comment: There are still changes required in the manuscript. To start with the study design as depicted by the appended flowchart misses the point for which I suggested why we needed the flowchart. The reason is, and I will make an example, did you have a cohort that neither smoked but worked in industries? The study must be clear. Even though smoking played no role (as per p-value), did you do associations as well? For example smoking and working in industry. Here, I present two situations that can confound the findings.
Response: Thank you for the comment. However, the study designed is a population-based analytical cross-sectional study and this has been changed in the manuscript. With regards to the flowchart and owing to then potential confounders such as their smoking status, and working history in industries, primarily necessitated the need to run for crude and adjusted odds ratios as shown in Table 3. The adjusted ORs has accounted for other confounding variables such as smoking status, working in industries, etc. as it shows the interactive effect/influence of our explanatory variables in the tables. The adjusted model had a better overall performance owing to the pseudo-R variables found in the results output. However, there is a need for case-control or even prospective cohort design to sufficiently quantify the risks in the exposed and non-exposed groups.
Comment: Again, the fact that Antenatal cards could not be found for some participants presents a limitation. This limitation really throws some of the findings into dispute. I refer to the contribution of infectious diseases.
Response: You are absolutely right that it constitutes a limitation for the study, and we have documented it in the limitation section of the manuscript.
Comment: If the study is a pilot study, then the title should reflect that.
Response: We have included this in the revised title. Thank you
